# The Predictive Value of Hyperuricemia on Renal Outcome after Contrast-Enhanced Computerized Tomography

**DOI:** 10.3390/jcm8071003

**Published:** 2019-07-10

**Authors:** Ming-Ju Wu, Shang-Feng Tsai, Cheng-Ting Lee, Chun-Yi Wu

**Affiliations:** 1Division of Nephrology, Department of Internal Medicine, Taichung Veterans General Hospital, Taichung 407, Taiwan; 2School of Medicine, Chung Shan Medical University, Taichung 402, Taiwan; 3Rong Hsing Research Center for Translational Medicine, Institute of Biomedical Science, College of Life Science, National Chung Hsing University, Taichung 402, Taiwan; 4Graduate Institute of Clinical Medical Science, School of Medicine, China Medical University, Taichung 404, Taiwan; 5Department of Life Science, Tunghai University, Taichung 407, Taiwan; 6School of Medicine, National Yang-Ming University, Taipei 112, Taiwan

**Keywords:** uric acid, contrast media, acute kidney injury, hemodialysis, chronic kidney disease

## Abstract

The aim of this study was to determine whether elevated serum level of uric acid (sUA) could predict renal outcome after contrast-enhanced computerized tomography (CCT). We used a historical cohort of 58,106 non-dialysis adult patients who received non-ionic iso-osmolar CCT from 1 June 2008 to 31 March 2015 to evaluate the association of sUA and renal outcome. The exclusion criteria were patients with pre-existing acute kidney injury (AKI), multiple exposure, non-standard volume of contrast, and missing data for analysis. A total of 1440 patients were enrolled. Post-contrast-AKI (PC-AKI), defined by the increase in serum creatinine ≥ 0.3 mg/dL within 48 h or ≥50% within seven days after CCT, occurred in 180 (12.5%) patients and the need of hemodialysis within 30 days developed in 90 (6.3%) patients, both incidences were increased in patients with higher sUA. sUA ≥ 8.0 mg/dL was associated with an increased risk of PC-AKI (odds ratio (OR) of 2.62; 95% confidence interval (CI), 1.27~5.38, *p* = 0.009) and the need of hemodialysis (OR, 5.40; 95% CI, 1.39~21.04, *p* = 0.015). Comparing with sUA < 8.0 mg/dL, patients with sUA ≥ 8.0 mg/dL had higher incidence of PC-AKI (16.7% vs. 11.1%, *p* = 0.012) and higher incidence of hemodialysis (12.1% vs. 4.3%, *p* < 0.001). We concluded that sUA ≥ 8.0 mg/dL is associated with worse renal outcome after CCT. We suggest that hyperuricemia may have potential as an independent risk factor for PC-AKI in patients receiving contrast-enhanced image study.

## 1. Introduction

Post-contrast acute kidney injury (PC-AKI) is one of the most common causes of acute kidney injury (AKI), independently associated with both morbidity and mortality [1,2,3,4,5,6]. Early awareness of the risk factors to eliminate the potentially preventable AKI after contrast-enhanced image studies is a critical healthcare issue [7]. Although estimated glomerular filtration rate (eGFR) has been widely accepted to detect the high-risk patients of AKI after contrast-enhanced image studies, there are several episodes of PC-AKI that have developed in patients without advanced chronic kidney disease (CKD). Thus, further studies are needed to identify more risk factors for predicting PC-AKI. 

Hyperuricemia has been linked to AKI and progression of CKD via both crystal-dependent and crystal-independent mechanisms [8]. Both urate and calcium phosphate crystals could induce oxidative stress and the expression of chemokine, and lead to the renal tubular epithelium and acute alterations in auto-regulation of renal blood flow which contribute to the decrease of renal perfusion and the subsequent injury of renal tubule [8,9,10,11]. Elevated serum level of uric acid has been reported as a novel marker for predicting AKI and mortality in several clinical settings, such as admission, percutaneous coronary intervention, and surgery, especially cardiovascular survey [12,13,14,15,16,17,18,19]. 

The traditional risk factors for predicting contrast-induced nephropathy include pre-existing renal disease, elderly people, diabetes mellitus, congestive heart failure, hypovolemic status, administration of nephrotoxic agents, or a large amount of contrast medium [20,21]. Metabolic syndrome and pre-diabetes have been proposed as new risk factors for contrast–induced nephropathy [22].

However, the predictive value of serum level of uric acid on the risk of PC-AKI after contrast-enhanced computerized tomography (CCT) has not been examined. The aim of this study was to determine whether serum uric acid could predict the risk of developing AKI and the need of dialysis after CCT, as well as the impact of PC-AKI on long-term change of renal function.

## 2. Patients and Methods

### 2.1. Study Design and Clinical Data Retrieval

We used a history cohort of 58,106 non-dialysis adult patients who received non-ionic iso-osmolar contrast, iodixanol (visipaque, Chicago, IL, USA), enhanced CT from 1 June 2008 to 31 March 2015 and had available a baseline serum level of creatinine and uric acid within two weeks before CCT to evaluate the association of serum uric acid and renal outcome. The exclusion criteria were patients with pre-existing AKI, recent exposure to contrast media within 30 days, volume of contrast medium ≠ 100cc (regular contrast volume for CCT), missing baseline serum creatinine, missing baseline serum uric acid (data within two weeks before CCT), and missing post-contrast serum creatinine within one week after CCT. The informed consent was waived because the study is on the basis of data collection from routine care. The institute review board of Taichung Veterans General Hospital approved this study (IRB TCVGH No: CE16164B). 

There was no formal protocol for the prevention of contrast-induced nephropathy at this hospital at the time of this study. We calculated the eGFR using the four variables chronic kidney disease epidemiology collaboration (CKD-EPI) equation [23].

To find the cutoff values of baseline serum uric acid potentially associated with renal outcome, patients were classified into five groups stratified by baseline serum uric acid level: ≤3.9, 4–5.9, 6–7.9, 8–9.9, ≥10 mg/dL.

### 2.2. Outcome Variables

The renal outcome was determined by the primary and secondary endpoints. The primary renal endpoint was PC-AKI, which is defined by absolute increase of serum creatinine ≥0.3 mg/dL from baseline within 48 h or ≥50% within seven days after CCT, the Kidney Disease Improving Global Outcomes (KDIGO) criteria of AKI [1]. We did not include the criteria of urine volume of KDIGO criteria of AKI, because we could not collect enough data of urine volume in our study cohort. The secondary endpoint studied was the need of emergent hemodialysis after CCT. We identified the first procedure of hemodialysis within 30 days after CCT as PC-AKI requiring emergent hemodialysis. We also examined the differences in patient’s characteristics, clinical factors, and incidence of AKI after CCT between serum uric acid ≥8.0 mg/dL and <8.0 mg/dL.

### 2.3. Statistical Analysis

All statistical analyses were performed using the SPSS software (Statistical Package for the Social Science, version 20.0, Armonk, NY, USA). Quantitative data are expressed as mean ± standard deviation. Nominal and categorical variables were compared using the chi-squared likelihood ratio or Fisher exact test with post-hoc analyses to detect difference between each pair with bonferroni. Continuous variables were compared using the nonparametric Wilcoxon test. Stepwise multivariate logistic regression analysis was used to examine the independent association of PC-AKI with patient-related characteristics and clinical factors. Association between serum uric acid ≥8.0 mg/dL and the risk of acute kidney injury and dialysis within 30 days after contrast-enhanced computerized tomography was calculated by odds ratio (OR) and 95% confidence interval (CI). A two-sided *p* value of <0.05 was set to represent the statistical significance.

## 3. Results

### 3.1. Study Population

A total of 1440 eligible patients who received CCT were enrolled in the final study cohort (Figure 1). The age of study subjects ranged from 20 to 98 years (mean age 66.2 ± 15.7 years) and 66.9% patients were male. Among them, 354 (24.6%) participants had serum uric acid ≥8 mg/dL and 865 (60.1%) participants had eGFR greater than 60 mL/min/1.73 m^2^. Mean serum level of uric acid was 6.3 ± 2.7 mg/dL, mean serum creatinine level was 1.7 ± 2.1 mg/dL, and mean eGFR was 75.9 ± 48.0 ml/min/1.73 m^2^. The average times of measurement of baseline serum uric acid and serum creatinine were 6.4 ± 3.2 days and 5.2 ± 3.7 days before CCT. The high incidence of comorbidities was observed and listed in Table 1. Four subgroups were created after stratification by baseline serum levels of uric acid. There were 270, 430, 386, 225, and 129 patients in the groups of serum uric acid ≤3.9, 4–5.9, 6–7.9, 8–9.9, and ≥10 mg/dL, respectively. Higher baseline serum uric acid was associated with higher prevalence of old age, stage 3~5 CKD, hypertension, coronary arterial disease, heart failure, atrial fibrillation, and chronic liver disease (Table 1). 

### 3.2. Renal Outcome Rates after CCT

In total, 180 (12.5%) patients developed PC-AKI, the primary endpoint, and 90 (6.3%) patients received emergent hemodialysis within 30 days. Not unexpectedly, the incidence of PC-AKI increased from 7.1% in stage 1 CKD to 29% in stage 4 CKD (Figure 2A, *p* < 0.001), but decreased to 17.2 in stage 5 CKD. The incidence of emergent hemodialysis within 30 days after CCT increased from 1.8% in stage 1 CKD to 38.8% in stage 5 CKD (Figure 2B, *p* < 0.001). The Chi-square tests were used to detect the significant differences of PC-AKI and emergent hemodialysis within 30 dyas after CCT among all pairs of populations with different stages of chronic kidney disease, both *p* < 0.001. Result of post-hoc analyses are shown in Figure 2. 

Moreover, the incidence of PC-AKI decreased in lower ranges of serum uric acid, from 17.8% in patients with serum uric acid ≥10 mg/dL to 8.1% in patients with serum uric acid 4–5.9 mg/dL (Figure 3A, *p* < 0.001), but increased to 12.2% in patients with serum uric acid ≤3.9 mg/dL. There was a J-shaped relationship between serum uric acid and PC-AKI after CCT. The incidence of emergent hemodialysis within 30 days after CCT increased from 2.2% in patients with serum uric acid <4 mg/dL to 15.5% in patients with serum uric acid ≥ 10 mg/dL (Figure 3B, *p* < 0.001). The Chi-square tests were used to detect the significant differences among all pairs of populations with different ranges of serum uric acid, both *p* < 0.001. Result of post-hoc analyses are shown in Figure 3.

### 3.3. Sensitivity Analysis: Impact of Serum Acid on Renal Outcome after CCT

Table 2 shows that the baseline serum uric acid ≥8.0 mg/dL was significantly associated with PC-AKI (OR, 1.54; 95% CI, 1.10~2.18, *p* = 0.013) and emergent hemodialysis within 30 days after CCT (OR, 2.93; 95% CI, 1.90~4.52, *p* < 0.001). Serum uric acid remained associated with PC-AKI (OR, 2.62, 95% CI, 1.27~5.38, *p* = 0.009) and emergent hemodialysis within 30 days after CCT (OR, 5.40, 95% CI, 1.39~21.04, *p* = 0.015) after adjustment for the age, gender, comorbidities (cancer, diabetic mellitus, hypertension, coronary arterial disease, heart failure, atrial fibrillation, cerebral vascular attack, chronic liver disease, peripheral arterial occlusive disease, gastrointestinal bleeding, and shock) and baseline laboratory data (serum albumin, hemoglobin) and medications (diuretics, ACEi/ARB, N-acetylcyestine, sodium bicarbonate, NSAID).

Incidence of serum uric acid ≥8.0 mg/dL significantly increased as the progression of renal function, from 6.2% in stage 1 CKD to 52% in stage 4 CKD, and decreased slightly to 41.4% in stage 5 CKD (Figure 4A, *p* < 0.001). Notably, in patients with stage 1 and 2 CKD, but not in patients with stage 3~5 CKD, serum uric acid ≥8.0 mg/dL was significantly associated with higher incidence of PC-AKI when comparing with serum uric acid < 8.0 mg/dL (Figure 4B, *p* < 0.05 in both stage 1 and stage 2 CKD). Patients with serum uric acid ≥ 8.0 mg/dL had higher incidence of emergent hemodialysis within 30 days after CCT in stage 2, 3A, and 3B CKD (Figure 4C). Overall, when comparing with serum uric acid < 8.0 mg/dL, patients with serum uric acid ≥ 8.0 mg/dL had higher incidence of PC-AKI (16.7% vs. 11.1%, *p* = 0.012, Figure 4D) and higher incidence of emergent hemodialysis within 30 days after CCT (12.1% vs. 4.3%, *p* < 0.001, Figure 4D). Compared to male patients, female patients had significantly higher risk to receive hemodialysis within 30 days after CCT (8.1% vs. 4.7%, *p* = 0.012), but not in PC-AKI (15.9% vs. 12.3%, *p* = 0.075). However, gender is not an independent risk factor when we perform regression analysis to detect the potential risk factor of post-contrast AKI.

### 3.4. Analysis of Renal Outcome in three Months after CCT

We further collected renal function three months after CCT to compare the change of eGFR between AKI and non-AKI groups. The mean eGFR decreased 19.6 ± 37.4% in the AKI group and increased 1.3 ± 36.0% in the non-AKI group (*p* < 0.001, Figure 5A). Among them, 53.8% of patients with AKI had eGFR decreased by ≥20% compared to only 25.9% in patients without AKI (*p* < 0.001, Figure 5B). 

### 3.5. Subgroups Analysis of Renal Outcome after CCT

In the subgroups analysis, we found the incidences of serum uric acid ≥ 8.0 mg/dL were 24.7% of 984 patients aged ≥60 years, 19.7% of 476 female patients, 26.4% of 870 hypertensive patients, and 24.3% of 452 patients with cancer. The odds ratios of serum uric acid ≥8.0 mg/dL for the predicting PC-AKI were 2.61 (95% CI = 1.02~2.24, *p* = 0.040) in patients aged ≥60 years, and 2.19 (95% CI = 1.21~3.94, *p* = 0.009) in female patients. The odds ratios of serum uric acid ≥8.0 mg/dL for the predicting emergent hemodialysis within 30 days after CCT were 2.07 (95% CI = 1.23~3.49, *p* = 0.006) in patients aged ≥60 years, 7.03 (95% CI = 3.01~16.44, *p* < 0.0001) in patients aged <60 years, 2.68 (95% CI = 1.02~5.89, *p* = 0.014) in female patients, 3.06 (95% CI = 1.81~5.17, *p* < 0.0001) in male patients, 3.00 (95% CI = 1.82~4.94, *p* < 0.0001) in hypertensive patients, 2.43 (95% CI = 1.34~4.41, *p* = 0.003) in diabetic patients, 3.49 (95% CI = 1.82~6.66, *p* < 0.0001) in non-diabetic patients, 2.73 (95% CI = 1.24~6.01, *p* = 0.013) in patients with coronary arterial disease, and 3.00 (95% CI = 1.78~5.05, *p* < 0.0001) in patients without coronary arterial disease.

## 4. Discussion

The primary finding of this study is the strong association between hyperuricemia and the risk of PC-AKI and the need of emergent hemodialysis within 30 days after CCT. Even after adjustment for patient characteristics, comorbidities, laboratory data, and medications, pre-contrast serum uric acid continued to be strongly associated with renal outcome after CCT. Our findings provide proof of concept that hyperuricemia, especially when serum uric acid is ≥8.0 mg/dL, was associated with higher risk of PC-AKI after CCT. The association between hyperuricemia and PC-AKI occurs more significantly in patients without advanced CKD stage 4 and 5.

The estimated GFR has been widely used to assess the risk of PC-AKI when patients need to receive contrast-enhanced image studies [2]. Regardless of the fact that hyperuricemia is more common in patients with advanced CKD, 11.6% of 865 patients with stage 1 and 2 CKD had serum uric acid ≥8.0 mg/dL in this study. These patients without advanced stage 3~5 CKD are generally considered to have relatively lower risk of PC-AKI. However, the incidence of PC-AKI and emergent hemodialysis within 30 days after CCT were 9.1% and 1.8%, respectively, in patients with eGFR ≥ 60 mL/min/1.73 m^2^. In this study, we suggest that hyperuricemia could be one of the independent risk factors for the prediction of renal outcome after CCT. The impact of PC-AKI was further demonstrated by the fact that a significantly higher percentage of eGFR decreased ≥20% after three months of CCT. 

The possible explanations for the increased risk of PC-AKI in patients with elevated serum uric acid include both crystal-dependent and crystal-independent mechanisms [8,24]. Elevated serum uric acid can induce renal vasoconstriction and impair auto-regulation, which leads to reduced renal blood flow and GFR [9,10]. A mild elevation of serum uric acid in rats could cause renal vasoconstriction in a crystal-independent pathway [11]. Several recent studies demonstrated that hyperuricemia could worsen the injury of the renal tubule via pro-inflammatory pathways involving activation of the renin-angiotensin system, chemokine expression, and endothelial dysfunction [8,9]. Importantly, contrast medium may increase the burden of hyperuricemia-induced kidney injury.

The association between hyperuricemia and an increased risk for developing AKI has been demonstrated in patients receiving cardiovascular surgery and percutaneous coronary interventions and acute paraquat intoxication [12,13,14,15,16,17,18,19]. Moreover, hyperuricemia has been proposed as a novel marker for early detection of AKI [25]. In this study, we demonstrate that hyperuricemia is an important predictor of developing PC-AKI and the need for emergent hemodialysis within 30 days after CCT.

Interestingly, the association between serum uric acid ≥ 8.0 mg/dL and PC-AKI was more significant in patients with stage 1 and 2 CKD, which accounts for 60.1% in this study cohort. Recently, Kuwabara and colleagues also reported that change in SUA is independently associated with change in eGFR over time in patients with eGFR ≥ 60 mL/min/1.73 m^2^ [26]. These findings suggest that the impact of hyperuricemia, sUA ≥ 8.0 mg/dL, on PC-AKI is more prominent in early stage CKD patients. However, we do not have enough sUA after contrast CT to evaluate if the change of sUA will also impact on the development of PC-AKI. 

On the other hand, female patients have significantly higher risk of receiving hemodialysis within 30 days after CCT, but not in PC-AKI. In general, the female population has lower average sUA than the male population. Although our results could not support the female gender as an independent risk factor to predict PC-AKI, more study is necessary to clarify the gender effect in the association of sUA and PC-AKI.

Even though increasing evidence support the idea that hyperuricemia may increase the risk of AKI development, interventions by lowering serum level of uric acid to prevent AKI remain scarce. A small-scale randomized control study showed that lowering serum uric acid by rasburicase, an urate oxidase, did not reduce the development of AKI after cardiac surgery by using traditional and non-traditional markers [27]. It is worth mentioning that hyperuricemia could be a reflection of diminished renal perfusion. Prerenal azotemia may lead to enhanced proximal tubular reabsorption of salt, water, urea, as well as uric acid [28].

In support of our findings, Lapsia and coworker demonstrated a J-shaped relationship between hyperuricemia and postoperative AKI [12]. An explanation for the higher risk of AKI in patients with lower serum levels of uric acid, <4 mg/dL for example, is due to oxidative stress, as uric acid can act as both an anti-oxidant and pro-oxidant agent [29,30]. Malnutrition and inflammation were also suggested to be important factors for lower serum levels of uric acid and a worse outcome [19]. Moreover, the systematic review and meta-analysis among the patients undergoing coronary angiography and/or percutaneous coronary intervention showed that hyperuricemia is independently associated with the occurrence of contrast–induced AKI and the risk of renal replacement therapy [31].

There are several important limitations in this study. This is a single-center historic cohort study. Out of 58,106 study subjects, only 1440 patients were included in analysis. The result is subject to selection bias and the finding might have limited generalizability. The statistical power was limited to detect the impact of hyperuricemia in AKI requiring dialysis in eGFR ≥ 60 mL/min/1.73 m^2^ (*n* = 16, 1.8%) and serum uric acid < 4 mg/dL (*n* = 6, 2.2%), which is an important clinical end point. On the other hand, there is also no data available to evaluate if the intervention of lower serum uric acid will reduce the risk of PC-AKI. Since this study is an observational study in nature, it is difficult to show the causality. We do not have renal biopsy data to confirm the cause of PC-AKI. A multi-center, prospective large-scale study is eventually required to address these limitations.

## 5. Conclusions

Our findings provide additional evidence to demonstrate that elevated serum uric acid is an independent risk factor for AKI in patients undergoing contrast-enhanced image study PC-AKI. Moreover, we provide further evidence that PC-AKI is associated with the need of dialysis and long-term renal function progression. We suggest that serum level of uric acid, together with eGFR, is necessary for patients scheduled to receive CCT.

## Figures and Tables

**Figure 1 jcm-08-01003-f001:**
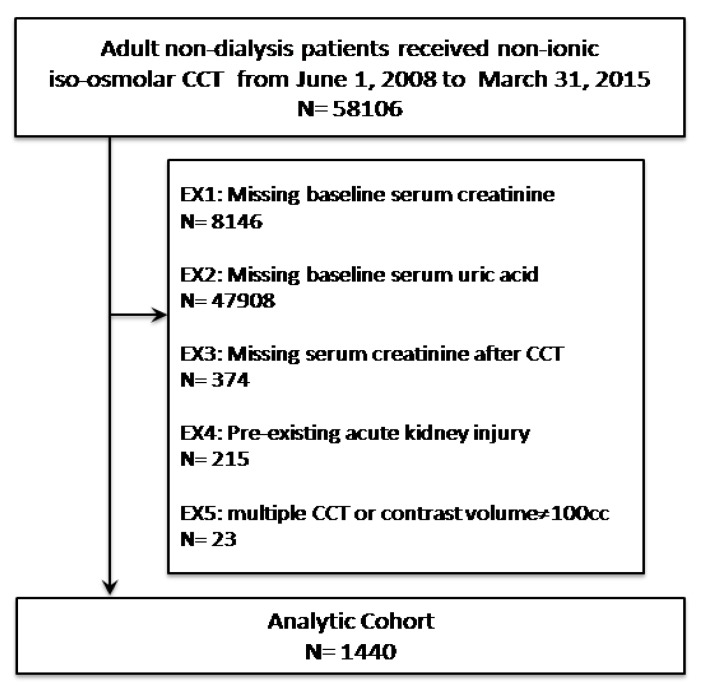
Study population selection flow diagram. CCT = Contrast-enhanced computerized tomography. EX = Exclusion.

**Figure 2 jcm-08-01003-f002:**
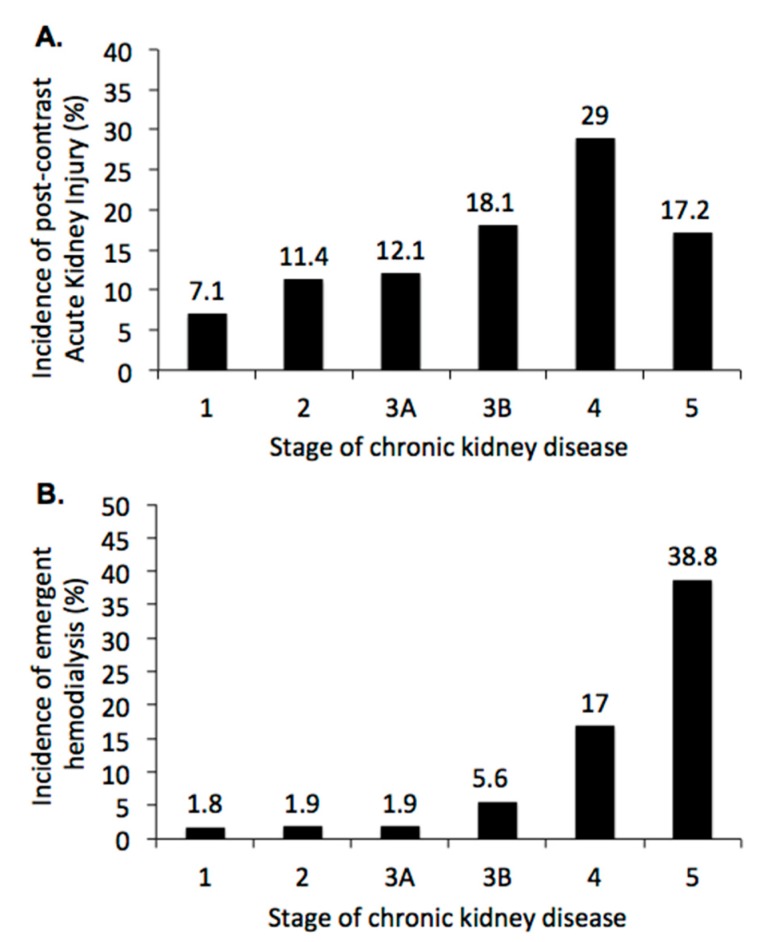
Renal outcome after contrast-enhanced computerized tomography (CCT) by the stage of chronic kidney disease: (**A**) Incidence of post-contrast acute kidney injury, defined by absolute increase of serum creatinine ≥0.3 mg/dL from baseline within 48 h or ≥50% within seven days after CCT. Chi-square tests were used to detect the significant differences among all pairs of populations with different stages of chronic kidney disease, *p* < 0.001. Post-hoc analysis showed *p* = 0.004 in stage 1 vs. 3B, *p* = 0.026 in stage 1 vs. 4, *p* = 0.001 in stage 1 vs. 5, *p* = 0.001 in stage 2 vs. 4, *p* = 0.006 in stage 3A vs. 4. (**B**) Incidence of emergent hemodialysis within 30 days after CCT. Chi-square tests were used to to detect the significant differences among all pairs of populations with different stages of chronic kidney disease, *p* < 0.001. Post-hoc analysis showed *p* < 0.001 in stage 1 vs. 4, *p* < 0.001 in stage 1 vs. 5, *p* < 0.001 in stage 2 vs. 4, *p* < 0.001 in stage 2 vs. 5, *p* < 0.001 in stage 3A vs. 4, *p* < 0.001 in stage 3A vs. 5, *p* < 0.001 in stage 3B vs. 5, *p* < 0.007 in stage 4 vs. 5, respectively.

**Figure 3 jcm-08-01003-f003:**
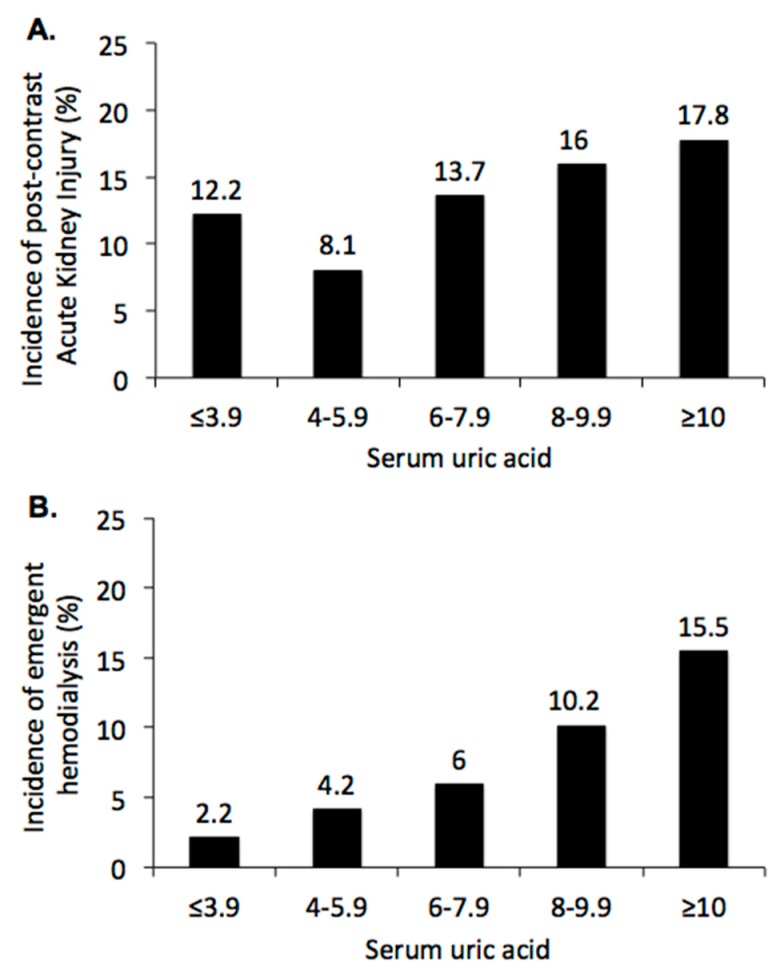
Renal outcome after contrast-enhanced computerized tomography stratified by baseline serum uric acid: (**A**) Incidence of post-contrast acute kidney injury, defined by absolute increase of serum creatinine ≥0.3 mg/dL from baseline within 48 h or ≥50% within seven days after CCT. Chi-square tests were used to detect the significant differences among all pairs of populations with different ranges of serum uric acid, *p* < 0.001. Post-hoc analysis showed *p* = 0.033 in sUA 4–5.9 vs. 8–9.9 mg/dL, *p* = 0.028 in sUA 4–5.9 vs. ≥10 mg/dL, respectively. (**B**) Incidence of emergent hemodialysis within 30 days after contrast enhanced computerized tomography. Chi-square tests were used to detect the significant differences among all pairs of populations with different ranges of serum uric acid, *p* < 0.001. Post-hoc analysis showed *p* = 0.002 in sUA ≤ 3.9 vs. 8–9.9 mg/dL, *p* < 0.001 in sUA ≤ 3.9 vs. ≥10 mg/dL, *p* = 0.036 in sUA ≤ 3.9 vs. 6–9.9 mg/dL, *p* < 0.001 in sUA 4–5.9 vs. ≥10 mg/dL, *p* = 0.015 in sUA 6–7.9 vs. ≥10 mg/dL, respectively.

**Figure 4 jcm-08-01003-f004:**
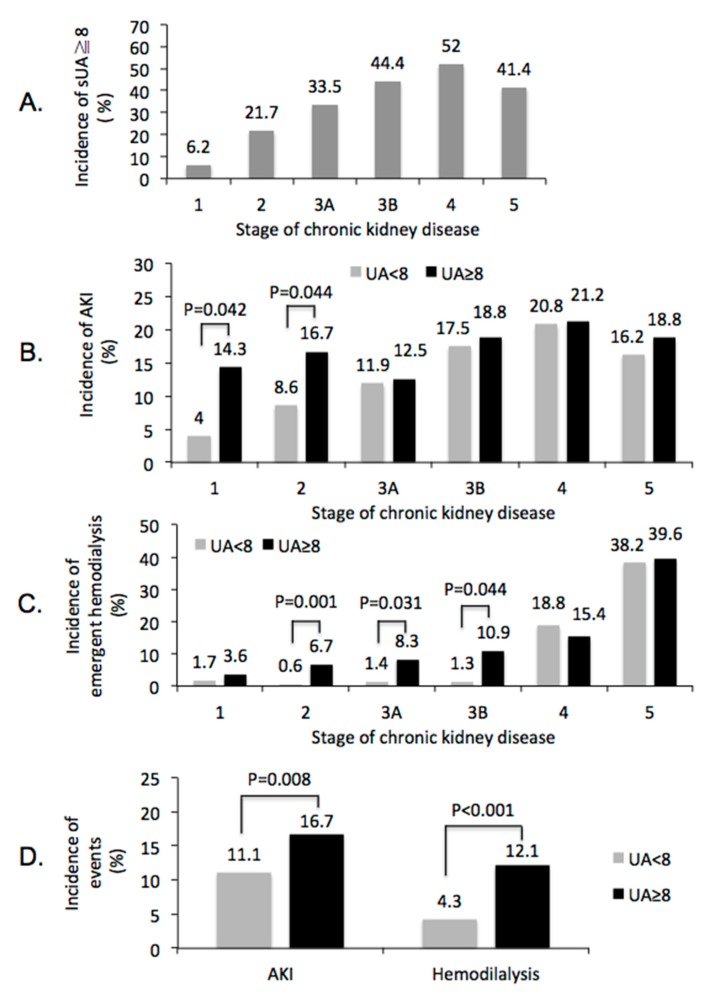
Impact of sUA ≥8.0 mg/dL on renal outcome after contrast-enhanced computerized tomography. (**A**) percentages of patients with sUA ≥ 8 mg/dL by stage of CKD, Chi-square tests were used to detect the significant differences among all pairs of populations with different stages of CKD, *p* < 0.001. (**B**) incidence of post-contrast AKI by stage of CKD. (**C**) Incidence of emergent hemodialysis within 30 days by stage of CKD. (**D**) difference of renal events, post-contrast AKI, and emergent hemodialysis with 30 days after contrast-enhanced computerized tomography, between patients with sUA <8 and ≥8 mg/dL. sUA = Serum uric acid. AKI = Acute kidney injury. CKD = Chronic kidney disease.

**Figure 5 jcm-08-01003-f005:**
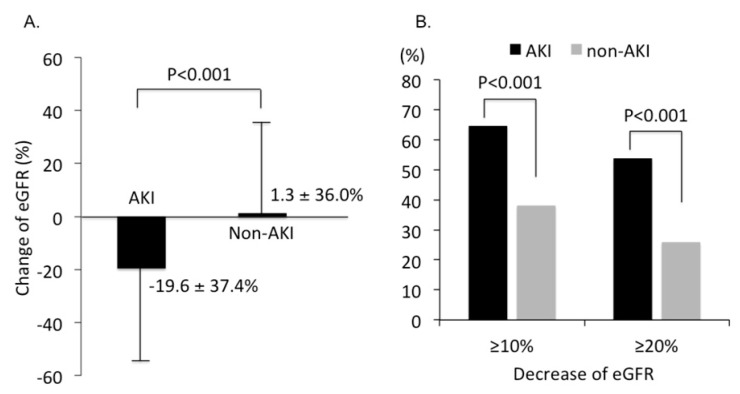
Impact of post-contrast acute kidney injury on renal outcome at three months after contrast-enhanced computerized tomography. (**A**) Change of eGFR at three months after contrast-enhanced computerized tomography. (**B**) Percentage of patients with eGFR decreased by ≥10% and ≥20% in patients with or without AKI, respectively. sUA = Serum uric acid. AKI = Acute kidney injury. eGFR = Estimated glomerular filtration rate.

**Table 1 jcm-08-01003-t001:** Baseline characteristics.

Serum Uric Acid (Number)	≤3.9 (N = 270)	4.0–5.9 (N = 430)	6.0–7.9 (N = 386)	8–9.9 (N = 225)	≥10 (N = 129)	Total (N = 1440)	*p* Value
**Clinical characteristics**							
Age (years)	63.2 ± 17.0	65.2 ± 15.5	67.5 ± 14.5	68.8 ± 14.5	67.5 ± 16.9	66.2 ± 15.7	<0.0001
Age ≥ 65 years *	131 (48.5%)	229 (53.3%)	244 (63.2%)	141 (62.7%)	76 (59%)	821 (57%)	<0.0001
Male sex *	176 (65.2%)	279 (64.9%)	254 (65.8%)	162 (72%)	93 (72.1%)	964 (66.9%)	0.233
**Status of renal function**							
Stage 1 CKD	175 (64.8%)	177 (41.2%)	71 (18.4%)	20 (8.9%)	8 (6.2%)	451 (31.3%)	<0.0001
Stage 2 CKD	56 (20.7%)	135 (31.4%)	133 (34.5%)	60 (26.7%)	30 (23.3%)	414 (28.8%)	
Stage 3A CKD	14 (5.2%)	57 (13.3%)	72 (18.7%)	53 (23.6%)	19 (14.7%)	215 (14.9%)	
Stage 3B CKD	10 (3.7%)	27 (6.3%)	43 (11.1%)	31 (13.8%)	33 (25.6%)	144 (10%)	
Stage 4 CKD	4 (1.5%)	14 (3.3%)	30 (7.8%)	29 (12.9%)	23 (17.8%)	100 (6.9%)	
Stage 5 CKD	11 (4.1%)	20 (4.7%)	37 (9.6%)	32 (14.2%)	16 (12.4%)	116 (8.1%)	
**Comorbidity**							
Cancer *	82 (30.4%)	141 (32.8%)	115 (29.8%)	77 (34.2%)	37 (28.7%)	452 (31.4%)	0.689
Diabetic mellitus *	93 (34.4%)	153 (35.6%)	132 (34.2%)	87 (38.7%)	49 (38%)	514 (35.7%)	0.786
Hypertension *	125 (46.3%)	233 (54.2%)	271 (70.2%)	154 (68.4%)	87 (67.4%)	870 (60.4%)	<0.0001
CAD *	48 (17.8%)	107 (24.9%)	127 (32.9%)	60 (26.7%)	47 (36.4%)	389 (27%)	<0.0001
Heart failure *	15 (5.6%)	30 (7%)	28 (7.3%)	14 (6.2%)	23 (17.8%)	110 (7.6%)	<0.0001
Atrial fibrillation *	27 (10%)	45 (10.5%)	59 (15.3%)	25 (11.1%)	35 (27.1%)	191 (13.3%)	<0.0001
CVA *	43 (15.9%)	100 (23.3%)	87 (22.5%)	47 (20.9%)	27 (20.9%)	304 (21.1%)	0.197
Chronic liver disease *	21 (7.8%)	34 (7.9%)	32 (8.3%)	24 (10.7%)	21 (16.3%)	132 (9.2%)	0.036
PAOD *	5 (1.9%)	19 (4.4%)	24 (6.2%)	14 (6.2%)	9 (7%)	71 (4.9%)	0.061
Shock *	30 (11.1%)	18 (4.2%)	14 (3.6%)	11 (4.9%)	14 (10.9%)	87 (6%)	<0.0001
GI bleeding *	12 (4.4%)	9 (2.1%)	15 (3.9%)	11 (4.9%)	9 (7%)	56 (3.9%)	0.098
**Laboratory data**							
Serum Albumin	3.0 ± 0.6	3.4 ± 0.8	3.4 ± 0.7	3.4 ± 0.7	3.3 ± 0.7	3.3 ± 0.7	<0.0001
Hemoglobin	10.3 ± 2.4	11.5 ± 2.6	11.3 ± 2.6	11.3 ± 2.6	10.8 ± 2.6	11.1 ± 2.6	<0.0001

CAD: Coronary arterial disease; CVA: Cerebral vascular attack; PAOD: Peripheral arterial occlusive disease; GI bleeding: Gastrointestinal bleeding. Data are expressed as mean ± standard deviation unless otherwise stated. * Data are n (%).

**Table 2 jcm-08-01003-t002:** Association between serum uric acid ≥8.0 mg/dL and the risk of acute kidney injury and dialysis within 30 days after contrast-enhanced computerized tomography.

	Odd Ratio	95% Confident Interval	*p* Value
**Risk of acute kidney injury**			
Unadjusted	1.54	1.10~2.18	0.013
Adjusted, model 1	2.40	1.31~4.42	0.005
Adjusted, model 2	2.62	1.27~5.38	0.009
**Risk of dialysis within 30 days after CCT**			
Unadjusted	2.93	1.90~4.52	<0.0001
Adjusted, model 1	6.42	1.91~21.56	0.003
Adjusted, model 2	5.40	1.39~21.04	0.015

Definition of acute kidney injury is absolute increase of serum creatinine ≥0.3 mg/dL from baseline within 48 h or ≥50% within seven days after contrast-enhanced computerized tomography. Model 1, adjusted by the comorbidities listed in Table 1. Model 2, adjusted by hemoglobin, serum albumin, bilirubin, uric acid, usage of diuretics, usage of ACE inhibitors/ARB, usage of N-acetylcysteine, usage of sodium bicarbonate, usage of non-steroidal anti-inflammatories, plus covariates listed in Model 1.

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
