# Peer review of "The Predictive Value of Hyperuricemia on Renal Outcome after Contrast-Enhanced Computerized Tomography"

_jcm, 2019, doi:10.3390/jcm8071003_

Reviewer 1 Report

The study showed the positive association between hyperuricemia and the risk of post-contrast (PC)-acute kidney injury (AKI) and the need of emergent hemodialysis within 30 days after contrast-enhanced computerized tomography (CCT). Hyperuricemia (especially, serum uric acid ≥8.0 mg/dL) was associated with higher risk of post-contrast PC-AKI after CCT. The authors concluded that the findings demonstrate that elevated serum uric acid is an independent risk factor for AKI in patients undergoing CCT. The authors suggest that serum level of uric acid is necessary for patients scheduled to receive CCT.

This article is well-written and showed the importance of measuring serum level of uric acid as a risk factor for PC-AKI after CCT. I made some comments.

1.       I cannot understand where p<0.001 indicates in Figure 2, 3, and 4A. Which methods did the authors use in these analyses, one-way ANOVA or Welch? Does the p-value mean the difference only between stage 1 and stage 5? I would like to know the p value among each pairs. I suggest the authors to conduct ANOVA with post-hoc analyses to detect the difference between each pair, like Tukey or Bonferroni.

2.       Please add the reference of the definition of PC-AKI.

3.       An article showed that uric acid change is an independent risk for kidney disease (Am J Nephrology. 2017;45(4):330-337. PMID: 28285309). Could you assess uric acid change between pre and post CCT in this study? If you can, does increased uric acid cause a risk for AKI?

4.       I well understand that the authors adjusted the rate of male sex in Table1. However, I suggest the authors to conduct all the analyses by sex because serum uric acid level is completely different between male and female.

5.       This study is an observation study, and it is difficult to show the causality. I suggest the authors to add it in limitation.

This article showed important information that hyperuricemia is a risk factor for PC-AKI after CCT. We need further study whether uric acid lowering agent for hyperuricemia prevent PC-AKI after CCT. I am looking forward to reviewing your revised manuscript.

Author Response

Thank you very much for spending so much time to give us comments. Our study team members appreciate for your opinions to improve our manuscript. We have finished the point-by-point response to your comments. All the corrections have been marked with red color. 

1.      I cannot understand where p<0.001 indicates in Figure 2, 3, and 4A. Which methods did the authors use in these analyses, one-way ANOVA or Welch? Does the p-value mean the difference only between stage 1 and stage 5? I would like to know the p value among each pairs. I suggest the authors to conduct ANOVA with post-hoc analyses to detect the difference between each pair, like Tukey or Bonferroni.

Answer: Thank you for your valuable suggestion to improve our manuscript. The Chi-square tests were used to detect the significant differences among all pairs of populations in Figure 2,3 and 4A, all p<0.001. We perform the post-hoc analyses to detect difference between each pair with bonferroni as reviewer’s suggestion Result of post-hoc analyses were showed in figures. (Page 4, Line 132 to Page 6, Line 177; Page 8, Figure 4A and Line 225-227)

2.  Please add the reference of the definition of PC-AKI.

Answer: PC-AKI, which is defined by absolute increase of serum creatinine ³0.3 mg/dL from baseline within 48 hours or ³50% within 7 days after CCT, the KDIGO criteria of AKI. We did not include the criteria of urine volume of KDIGO criteria of AKI, because we could not collect enough data of urine volume in our study cohort. (Page 2, Line 80-81 and all other areas with definition of AKI)

3. An article showed that uric acid change is an independent risk for kidney disease (Am J Nephrology. 2017;45(4):330-337. PMID: 28285309). Could you assess uric acid change between pre and post CCT in this study? If you can, does increased uric acid cause a risk for AKI?

Answer: Thank you for the reviewer to remind us this limitation. Since serum uric acid is not regularly checked before and after contrast enhanced CT, there is no data available in this study to evaluate if the change of serum uric acid is associated with increased risk for AKI. We also add the information of this paper in our discussion and reference. Moreover, we have added this limitation in the discussion as your suggestion. (Page 10, Line 292-296)

4. I well understand that the authors adjusted the rate of male sex in Table1. However, I suggest the authors to conduct all the analyses by sex because serum uric acid level is completely different between male and female.

Answer: Thank you for reviewer’s opinion to suggest us conduct analyses the difference between male and female patients. About 2/3 of patients in this study were male and the ratio of male sex is similar in each ranges of sUA. Compared to male patients, female patients have significantly higher risk to receive hemodialysis within 30 days after CCT (8.1% vs. 4.7%, p=0.012), but not in PC-AKI (15.9% vs. 12.3%, p=0.075). However, gender is not an independent risk factor when we perform regression analysis to detect the potential risk factor of post-contrast AKI. (Page 7, line 217-220 and Page 10, Line 297-301)

5. This study is an observation study, and it is difficult to show the causality. I suggest the authors to add it in limitation. This article showed important information that hyperuricemia is a risk factor for PC-AKI after CCT. We need further study whether uric acid lowering agent for hyperuricemia prevent PC-AKI after CCT. I am looking forward to reviewing your revised manuscript.

Answer: Thank you for the reviewer to remind us this limitation. There is no data available to evaluate if the intervention to lower serum uric acid will reduce the risk of PC-AKI. Since this study is an observational study in nature, it is difficult to show the causality. A multicenter, prospective large-scale study is eventually required to address these limitations. We have added this limitation in the discussion as your suggestion. (Page 10, Line 323-325)

Reviewer 2 Report

1.       What is the definition of baseline serum uric acid. Must serum uric acid be measured within a certain time frame prior to contrast imaging

2.       What is the definition of baseline creatinine

3.       Why did you exclude patient who received volume of contrast ≠ 100 ml. Did it mean that all included patient had to receive 100 ml of contrast

4.       The uric acid category -<4, 4-6, 6-8, 8-10, ≥10” is not mutually exclusive. It is better to label as ≤3.9, 4-5.9, 6-7.9, 8-9.9, ≥10

5.       Please add reference for AK definition. Did you follow RIFLE, AKIN, or KDIGO criteria.

6.       Please provide detail of contrast medium in your hospital. Did you include on IV contrast imaging

7.       Out of 58106, only 1440 were included in analysis. The result is subject to selection bias and finding might have limited generalizability

8.       Table 2 showed association between serum uric acid of risk of AKI and dialysis. I am not sure if OR is per mg/dL increase in serum uric, or did you put uric acid as categorical variables in the model. However, if you put serum uric acid as continuous variable in the logistic model, the analysis might not be valid because based on Figure 3A, the association between serum uric acid and AKI is J-shape, not linear.

9.       Manuscript need English editing

Author Response

Thank you very much for spending so much time to give us comments. Our study team members appreciate for your opinions to improve our manuscript. We have finished the point-by-point response to your comments. All the corrections have been marked with red color.   

1.      What is the definition of baseline serum uric acid. Must serum uric acid be measured within a certain time frame prior to contrast imaging.

Answer: We use serum level of uric acid within 2 weeks before CCT as the baseline serum uric acid. The average times between the measurement of baseline serum uric acid and CCT was 6.4±3.2 days. (Page 2, Line 64)

2.       What is the definition of baseline creatinine?

Answer: We use serum level of creatinine within 2 weeks before CCT as the baseline serum creatinine. The average times between the measurement of baseline serum creatinine and CCT was 5.2±3.7 days. (Page 2, Line 64)

3.           Why did you exclude patient who received volume of contrast ≠ 100 ml. Did it mean that all included patient had to receive 100 ml of contrast?

Answer: The regular contrast medium volume for contrast enhanced CT is 100mL at this hospital. Thus we exclude 23 patients who received 50cc contrast medium. Most of them are advanced CKD patients. (Page 2, Line 67)

4.      The uric acid category -<4, 4-6, 6-8, 8-10, ≥10” is not mutually exclusive. It is better to label as ≤3.9, 4-5.9, 6-7.9, 8-9.9, ≥10.

Answer: Thank you for the opinion of reviewer, we change the uric acid category in all this manuscript as reviewer’s suggestion. (Page 2, Line 76)

5.         Please add reference for AK definition. Did you follow RIFLE, AKIN, or KDIGO criteria.

Answer: PC-AKI, which is defined by absolute increase of serum creatinine ³0.3 mg/dL from baseline within 48 hours or ³50% within 7 days after CCT, the KDIGO criteria of AKI. We did not include the criteria of urine volume of KDIGO criteria of AKI, because we could not collect enough data of urine volume in our study cohort. The reference was also added. (Page 2, Line 80-81 and all other areas with definition of AKI)

6.       Please provide detail of contrast medium in your hospital. Did you include on IV contrast imaging.

Answer: All of the study subjects enrolled in this study received intravenous iso-osmolar non-ionic dimeric contrast agent, Iodixanol (Visipaqueâ), for enhanced CT at this study site. (Page 2, Line 63)

7.       Out of 58106, only 1440 were included in analysis. The result is subject to selection bias and finding might have limited generalizability.

Answer: Thank you for the valuable opinion of reviewer about this limitation. Serum uric acid is not regularly checked before contrast enhanced CT in current clinical practice. We could only use those patients have serum uric acid within 2 weeks before contrast enhanced CT in this retrospective study. Based on this study, we recommend to routine check serum uric acid, like serum creatinine, before the contrast enhanced CT. Moreover, we add this limitation in the discussion. (Page 10, Line 319-320)

8.       Table 2 showed association between serum uric acid of risk of AKI and dialysis. I am not sure if OR is per mg/dL increase in serum uric, or did you put uric acid as categorical variables in the model. However, if you put serum uric acid as continuous variable in the logistic model, the analysis might not be valid because based on Figure 3A, the association between serum uric acid and AKI is J-shape, not linear.

Answer: Thank you for reviewers to point out this mistake in our manuscript. The association between serum uric acid ³8.0 mg/dL and the risk of acute kidney injury and dialysis within 30 days after contrast-enhanced computerized tomography was calculated by odds ratio (OR) and 95% confidence interval (CI). (Page 3, Line 95-98 and Page 7, Table 2)

9.       Manuscript need English editing

Answer: Our manuscript has been editing by expert. Proof of English editing has been attached. We will request English editing from MDPI if more English Change is required as your suggestion.

Round  2

Reviewer 1 Report

This revised article looks much better than before. I agree with the authors’ response. I have no more suggestion. I thank the authors for this nice revision. I am very happy that I could review this nice article. Thank you, again.

Reviewer 2 Report

all of my comments have been addressed. I have no further comments.